# Hydrolytic Mechanism of a Metalloenzyme Is Modified by the Nature of the Coordinated Metal Ion

**DOI:** 10.3390/molecules28145511

**Published:** 2023-07-19

**Authors:** Zeyad H. Nafaee, Bálint Hajdu, Éva Hunyadi-Gulyás, Béla Gyurcsik

**Affiliations:** 1Department of Inorganic, Organic and Analytical Chemistry, University of Szeged, Dóm tér 7, H-6720 Szeged, Hungary; n.zeyad@chem.u-szeged.hu (Z.H.N.); balinth11@chem.u-szeged.hu (B.H.); 2College of Pharmacy, University of Babylon, Hillah 51001, Iraq; 3Laboratory of Proteomics Research, Biological Research Centre, Eötvös Loránd Research Network, Temesvári krt. 62, H-6726 Szeged, Hungary; gulyas.eva@brc.hu

**Keywords:** colicin E7, metallonuclease, circular dichroism, mass spectrometry, gel electrophoresis

## Abstract

The nuclease domain of colicin E7 cleaves double-strand DNA non-specifically. Zn^2+^ ion was shown to be coordinated by the purified NColE7 as its native metal ion. Here, we study the structural and catalytic aspects of the interaction with Ni^2+^, Cu^2+^ and Cd^2+^ non-endogenous metal ions and the consequences of their competition with Zn^2+^ ions, using circular dichroism spectroscopy and intact protein mass spectrometry. An R447G mutant exerting decreased activity allowed for the detection of nuclease action against pUC119 plasmid DNA via agarose gel electrophoresis in the presence of comparable metal ion concentrations. It was shown that all of the added metal ions could bind to the apoprotein, resulting in a minor secondary structure change, but drastically shifting the charge distribution of the protein. Zn^2+^ ions could not be replaced by Ni^2+^, Cu^2+^ and Cd^2+^. The nuclease activity of the Ni^2+^-bound enzyme was extremely high in comparison with the other metal-bound forms, and could not be inhibited by the excess of Ni^2+^ ions. At the same time, this activity was significantly decreased in the presence of equivalent Zn^2+^, independent of the order of addition of each component of the mixture. We concluded that the Ni^2+^ ions promoted the DNA cleavage of the enzyme through a more efficient mechanism than the native Zn^2+^ ions, as they directly generate the nucleophilic OH^−^ ion.

## 1. Introduction

Colicins represent a class of bacterial toxins, and as such they are intriguing targets of antibiotic research [1,2]. They are classified based on the mode of import into the attacked cells or on the mechanism of the toxicity [3,4]. Colicin E7 is a metalloprotein, produced by *Escherichia coli* under stress conditions to protect the cells from related bacteria as part of the cell defence mechanism [5,6]. Colicin E7 consists of three domains: the receptor-binding domain recognises the target cell surface, the translocation domain facilitates traffic across the cell membrane and the nuclease domain NColE7 is cleaved off while entering the cell, and it digests the cell’s DNA non-specifically [4,6,7,8,9]. The Im7 immunity protein is co-expressed with colicin E7 in the host cell, protecting it from nuclease action by interacting with the DNA binding site of NColE7 [8] (Figure 1). NColE7 is thus expressed and purified together with the Im7 protein under laboratory conditions. The HNH active site of the NColE7 enzyme is located at the C-terminus and exhibits a characteristic ββα secondary structure. The H545 histidine side-chain is suggested to participate in the generation of the OH^–^ ion nucleophile by promoting the deprotonation of the catalytic water molecule [10,11]. A divalent metal ion is bound by H544, H569 and H573 imidazole side-chains (throughout the text we apply the original numbering related to the full-size colicin E7 protein for better comparison with the literature data) in the active centre. Several crystal structures demonstrated that the metal ion can also bind a water molecule [12,13,14], to a phosphate [10,12,15] or sulphate [12,16] anion or to the scissile phosphodiester bond of the substrate DNA [17,18] via the fourth binding site in the tetrahedral geometry. This binding site proved to show a high affinity for Zn^2+^ ions, which are the metal ions that can be detected in NColE7, as obtained from the cells [14,17,19]. Purification by immobilised Ni^2+^ ion chromatography through oligohistidine tags may result in the replacement of Zn^2+^, while the procedure to break up the bonding between NColE7 and Im7 by decreasing the pH to ~3.0 leads to the apoprotein as the final product after separation from Im7 and renaturation [20,21]. 

Through mutational study, we have previously shown that the catalytic site of NColE7 is pre-organised to form the optimal tetrahedral cavity for Zn^2+^ ion binding. The 25 residue long N-terminal loop of NColE7 without a specific secondary structure was examined via computational modelling [22]. The suggested W464A mutation was identified to exert a dramatic effect on Zn^2+^ ion binding, as well as on the protein structure [23]. However, the structure and Zn^2+^-affinity were restored both via Im7 and DNA binding. This also draws attention to the fact that the structures of NColE7 crystallised together with these interacting agents reflect the induced position, instead of the native one [24].

N-terminal amino acids, with the main emphasis on R447, have been shown to promote the catalytic activity of NColE7. Removing R447 decreases the catalytic activity to ~10% of that of NColE7 [19,25]. As can also be seen in the tertiary structure, the N-terminus of the protein becomes close to the catalytic site so that it may interact with the DNA substrate (Figure 1). Such a conserved arginine in the *Serratia marcescens* endonuclease family was supposed to play a role in the positioning and polarisation of the scissile phosphodiester, but may also be involved in transition-state stabilisation [26]. We have previously shown that the mutation of the R447 arginine to glycine in NColE7 did not significantly influence the solution structure and DNA binding property of the protein [20], but it is easier to study its catalytic activity than that of the very active NColE7 enzyme. The aim of this work was to investigate the ability of the KGNK protein mutant of NColE7 (Figure 1) to bind transition metal ions such as Ni^2+^, Cu^2+^ and Cd^2+^ ions in the presence and absence of Zn^2+^ [20], as well as the effect of various metal ions on the catalytic process. We expected that the non-endogenous metal ions may be applied for the fine tuning of the enzyme activity.

## 2. Results and Discussion

### 2.1. Structural Aspects of the Interaction of the KGNK Protein with Metal Ions as Observed via Circular Dichroism Spectroscopy

The interaction of the KGNK protein with various transition metal ions was monitored via circular dichroism (CD) spectroscopy in aqueous solutions. The selected metal ions were Zn^2+^ ion as a common catalytic metal ion in the hydrolytic enzymes, Ni^2+^ ion as the most common component of the immobilised metal ion affinity chromatography (IMAC) technique for protein purification, Cu^2+^ ion as a strong Lewis acid which could strongly compete for histidine side-chains with Zn^2+^ ions and Cd^2+^ ion as a d^10^ electron system analogue of Zn^2+^ ion.

The KGNK protein was purified via its glutathione-S-transferase (GST) fused form in the presence of the immunity protein Im7, as described earlier [20]. After the cleavage of the GST affinity tag, the resulting protein has a single strong metal ion binding site in the active centre. There is a remaining short amino acid sequence consisting of eight residues at the N-terminus of the protein after the cleavage. These residues do not affect the structure, the Zn^2+^ ion/DNA binding or the catalytic activity of the enzyme [20]. The Im7 protein was removed by decreasing the pH to 3.0 and separating the components using ion exchange chromatography. Under such conditions, the metal ion, originally bound to the enzyme during the expression, is also lost, likely due to both the structural changes in NColE7 and the competition of protons with the metal ion for the imidazole donor groups of histidines. The following buffer exchange to 20 mM HEPES (pH 7.7) recovered the functional structure of the protein. Thus, we expected to obtain the apoprotein as the product of this procedure. A fraction of the KGNK protein was incubated with 10 equivalents of EDTA, followed by a buffer exchange to 20 mM HEPES (pH 7.7) to remove the chelator. The CD spectra of enzyme before and after the above EDTA treatment were identical (Figure 2a), suggesting that the resulting protein was indeed the apoenzyme.

Previously, we have shown that the binding of Zn^2+^ ion to the apoenzyme causes a slight but significant change in the circular dichroism spectrum, reflected in a red shift of about 2 nm and accompanied by a small change in the intensity [20,24]. This change could be reproduced in the presence of one equivalent Zn^2+^ ion, while the addition of further Zn^2+^ ions did not affect the CD spectra (Figure 2b).

The addition of equivalent non-endogenous metal ions, such as Ni^2+^, Cu^2+^ or Cd^2+^ ions, to the apoprotein caused very similar changes in the CD spectra to those related to the effect of Zn^2+^ ion (Figure 3). The red shift of about 2 nm was observed independent of the applied metal ion concentration; the same changes were detected upon adding both one and three equivalents (compared to the protein) of metal ions. These changes reflect that all of the added metal ions could strongly bind to the KGNK active centre under the applied conditions, causing a minor change in the secondary structure of the protein. This might be surprising, since the structure of the HNH motif offers a preformed tetrahedral binding site, consisting of three histidines, which is optimal for Zn^2+^-ions. At the same time, Ni^2+^ ions usually favour octahedral geometry, while Cu^2+^ would favour square planar coordination geometry. Nevertheless, Ni^2+^ ions were shown to bind to the HNH motif with a K_d_ of ~1 µM [27]. The metal ion coordination could be reversed by the addition of EDTA in equivalent amounts compared to the metal ions in each case, yielding CD spectra identical to those of the apoenzyme.

It is worthwhile to mention that the addition of excess EDTA to the apoenzyme (Figure 2a) caused a similar red shift of the spectra to that of the metal ions. This might be attributed to the interaction between the negatively charged EDTA with the positively charged residues of the protein otherwise participating in the DNA binding. The interaction with anions was already demonstrated by the crystal structure of an NColE7 mutant, in which phosphate ions occupied the positively charged DNA binding sites [16]. An excess of phosphate ions can also cause a slight change in the CD spectrum of the apo-KGNK, unlike the large excess of chloride ions (Figure 2c). Inhibition by the immunity protein Im7 is also largely based on similar interactions between the acidic side-chains of Im7 and basic side-chains of NColE7 [14]. Nevertheless, these results demonstrate that the interaction with complex anions is not exclusively ionic, but also partially structural, e.g., hydrogen bonding.

### 2.2. Mass Spectrometric Monitoring of Metal Ion Binding to KGNK

The interaction of KGNK with metal ions was also investigated via mass spectrometry. Two main peaks were detected upon measuring the KGNK protein without the addition of metal ions: one with ~20% relative intensity related to that of the mono-metallated KGNK with Zn(II) and one with ~80% relative intensity assigned to the apoenzyme. Na^+^ and K^+^ adducts were also detected in the MS measurements (Figure 4a). This result suggested that apoenzyme at the applied low concentrations could easily acquire metal ions from buffers/reagents/containers/sample holders applied during the experiments, even if these were treated very carefully. The expected masses in comparison with the obtained ones are collected in Table 1.

Next, equivalent metal ions were added to the protein solution. It was observed that the apo KGNK coordinated the added Zn^2+^ ions, as well as the non-endogenous metal ions (Ni^2+^ and Cd^2+^) in the active centre. This resulted in the corresponding mono-metallated enzyme species according to the mass spectra in Figure 4b–d. In addition, at a molar ratio of 1:1, there was a small peak related to the Zn^2+^-bound KGNK protein.

This means that Ni^2+^ ions could bind to the apoprotein, but they could not replace the Zn^2+^ ions in the minor Zn^2+^–KGNK species. The formation of ternary complexes, including two different metal ions bound to the protein, could not be unequivocally excluded. The relative intensities of these peaks, to which the masses of such complexes could be assigned, were lower than 10%, and therefore their assignment was uncertain. Cd^2+^ ions proved to be weaker interacting agents than Ni^2+^ ions, since one equivalent of Cd^2+^ ions could not even saturate the available apoprotein fully, so that a small peak related to the uncomplexed KGNK was still detected after the addition of one equivalent of this metal ion.

Most probably, the Cu^2+^-bound protein was obtained upon the addition of equivalent Cu^2+^ ions (Figure 5). However, it was rather difficult to unequivocally prove the Cu^2+^ binding to KGNK via MS because of the small difference in the atomic weight, and as a consequence, in the molecular weight. For this reason, the isotope distribution pattern of the most abundant peak of the deconvoluted MH^+^ mass spectrum at 15,774 *m*/*z* was compared with the simulated isotope pattern of the Cu^2+^–KGNK and Zn^2+^–KGNK complexes. The observed pattern of the recorded spectrum aligned well with the expected pattern of the Cu^2+^ complex. Therefore, it can be concluded that Cu^2+^ can bind to the protein, in accordance with the CD results.

We also observed a drastic change in the charge distribution pattern in the mass spectra as a consequence of the addition of the metal salt solutions (Figure 6). The apoprotein possessed a wide range of charged species, with two intensity maxima at z = 16 and 10 positive charges.

A similar phenomenon was previously published with NColE2, E7, E8 and E9 proteins, which are closely related bacterial toxins, upon Zn^2+^ binding at pH = 7.2. It was suggested that the apoprotein may exist in two different conformations, where one is an open structure and the other is a more stable one [28,29]. The latter is stabilised in the presence of the Zn^2+^ ion. As Figure 6 shows, all of the added metal ions could shift the charge distribution toward the structure exhibiting lower charges.

To verify the ability of the applied metal ions to compete with Zn^2+^ ions for the HNH motif in NColE7, one equivalent of the Zn^2+^ ions was added to the protein before supplementing the non-endogenous metal ions.

Upon the addition of one equivalent of Ni^2+^, Cu^2+^ or Cd^2+^ ions to the Zn^2+^–protein complex, the peak of the Zn^2+^-bound protein was identified in each spectrum (Figure 7). Thus, none of the applied metal ions could compete with Zn^2+^ ions at a molar ratio of 1:1. This is in agreement with the fact that Zn^2+^ has a high affinity with the HNH motif, with K_d_ in the low nM range, as was suggested via the isothermal calorimetric titrations with NColE9 having an analogue HNH motif to NColE7 [27], as well as via similar investigations with NColE7 [20].

### 2.3. The Effect of Metal Ions on the Catalytic Activity of KGNK

The nuclease activity of KGNK was studied using pUC119 plasmid DNA as a substrate via agarose gel electrophoresis. Starting with a DNA substrate mainly containing the superhelical form of the plasmid, first, the open circular form appears as a consequence of single-strand nicks. Then, linear DNA is formed upon double-strand cleavage. This would be preferred over the open circular form in the case of dimerisation of the enzyme upon DNA binding. Finally, a series of non-specific cleavages of DNA results in the distribution of DNA fragments (manifested as a smear) shifting to smaller and smaller sizes with time. Figure 8a shows that the enzyme was able to cleave the DNA efficiently, even without adding the metal ion to it. The mass spectrometric measurements already demonstrated that the apoenzyme solution may contain some Zn^2+^ ions acquired from the trace amounts of this metal ion in the environment (buffers/reactants/containers, etc.). The DNA solution may also contain a residual concentration of Zn^2+^ ions. However, the amount of Zn^2+^-bound KGNK formed under these conditions should not cause a significant DNA cleavage. Previously, we have shown that the Zn^2+^-bound KGNK mutant has moderate activity against plasmid DNA [20]. This was also indicated by the addition of one equivalent of Zn^2+^ ions, resulting in a less active enzyme compared to the “apo” form. This would surprisingly suggest that the metal ion that allows for this significant nuclease activity is different from Zn^2+^, and it could not be detected via MS in such a small concentration. Interestingly, this activity could only slightly be inhibited by the addition of EDTA at up to 10 equivalents to the enzyme (results not shown). We suggest that the reason for this might be the competition between the positively charged amino acid side-chains for the negatively charged EDTA, as the interaction between EDTA and apo KGNK was already detected via CD spectroscopy (see above). Nevertheless, a large excess (>10×) of EDTA can inhibit the enzyme.

The Zn^2+^ ions added at three equivalents caused slight inhibition of the enzymatic process. Although Zn^2+^ ions were suggested to be essential for the nuclease activity of NColE7 [14,21], the excess of this metal ion was shown to have an inhibitory effect [21]. This might be attributed to the ability of the excess metal ion to prevent the H545-mediated OH^−^ generation via weak coordination to this histidine. Supplementing one equivalent of Cu^2+^ ions to the KGNK protein in the experiments, we observed similar behaviour to that of Zn^2+^ ions (Figure 8b). It is difficult to decide at this point whether the resulting low catalytic activity is due to the Cu^2+^-bound enzyme, or whether it is due to the remnant activity of the apoenzyme after the partial replacement of the supposed unknown metal ion. Nevertheless, the excess of Cu^2+^ ions further decreased the catalytic activity that could best be observed at 2 h incubation.

More efficient DNA cleavage is observed in the presence of Cd^2+^ ions than with Zn^2+^ or Cu^2+^ ions (Figure 8c). However, this activity is still less than that of the control experiment with the KGNK protein in the absence of added metal ions. Furthermore, the activity is further decreased with increasing amounts of Cd^2+^ ions. This is in agreement with the mass spectrometric findings about the weak binding of Cd^2+^ ions to the KGNK enzyme.

The catalytic experiments in the presence of Ni^2+^ ions demonstrated much higher activity with this metal ion than the apoprotein used as a control (Figure 8d vs. Figure 8a). It is even more intriguing that the excess of the Ni^2+^ ions did not significantly inhibit this highly active enzyme. The comparison of the time dependence of the band intensity related to the superhelical DNA form is shown in Figure 8e. These data indicate that although Zn^2+^ is more strongly bound to the KGNK protein than the other divalent metal ions by at least three orders of magnitude [27], and thus Ni^2+^ may not be the native metal ion, its presence still results in high activity of the enzyme. This suggests that Ni^2+^ ions promote the DNA cleavage of the enzyme via different mechanisms than those of the Zn^2+^-bound KGNK. While the nucleophilic OH^−^ ion generation mediated by H545 is not so efficient in this mutant than in NColE7, the Ni^2+^ ion may promote the deprotonation of a coordinated water molecule, which can serve as an attacking agent for DNA hydrolysis. The crystal structure of NColE7 with Ni^2+^ ion supports this assumption, showing that the metal ion is coordinated to a phosphate ion and a water molecule besides the three histidine side-chains in distorted trigonal bi-pyramidal geometry [17].

To verify that the DNA cleavage was induced by Ni^2+^ ions bound in the active site of the enzyme, we carried out competition experiments with Zn^2+^ ion. Independent of the order of addition of each component of the mixture, i.e., first adding Zn^2+^ and then Ni^2+^, or in the reverse order, identical observations were made (Figure 9a).

Accordingly, the catalytic activity substantially decreased in comparison to that of the Ni^2+^-bound enzyme. This is rather characteristic of the Zn^2+^–enzyme complex. This reflects the replacement of the Ni^2+^ ions by the Zn^2+^ ions due to the thermodynamics, i.e., due to the stronger binding of Zn^2+^ ions to the active centre. This is in accordance with the results from the mass spectrometric experiments carried out under competitive conditions. Figure 9b shows that Ni^2+^ ions cannot initiate DNA cleavage in the absence of the enzyme under the applied conditions. From these results, we can conclude that the R447G-modified NColE7 protein prefers to bind Zn^2+^ ion in its active centre, but it has much higher activity in its Ni^2+^ form, providing a chance for enzymatic regulation by the metal ion environment.

## 3. Materials and Methods

### 3.1. Reagents

All of the reagents were used as purchased, without further purification. Seakem LE agarose was obtained from Lonza, Rockland, ME, USA and LB medium, ZnCl_2_, NiCl_2_, CdCl_2_, CuCl_2_, disodium-EDTA, Na_2_HPO_4_ and NaH_2_PO_4_ were obtained from Reanal Ltd, Budapest, Hungary. NaCl (Szkarabeusz Laboratory, Chemical Industry and Trade Ltd., Pecs, Hungary) was used, and Acrylamide/bis-acrylamide (29/1) as a 30% (*w*/*v*) solution was from SERVA Electrophoresis GmbH, Heidelberg, Germany and VWR Chemicals, LLC; Solon, Ohio, USA. Bisacrylamide 2K (standard grade, extra pure) was bought from AppliChem Panreac, Darmstadt, Germany. KCl and tris(hydroxymethyl)aminomethane were purchased from Molar Chemicals Ltd., Halásztelek, Hungary. N-2-hydroxyethylpiperazine-N-2-ethane sulfonic acid (HEPES) and ammonium bicarbonate were from Sigma-Aldrich, St. Louis, MO, USA, glycine was from Duchefa Biochemie B. V., Haarlem, The Netherlands, tricine and SDS were from VWR Chemicals and isopropyl β-D-1-thiogalactopyranoside (IPTG, dioxane-free) was obtained from Thermo Scientific, while ampicillin sodium salt was purchased from Sigma-Aldrich. The pGEX-6P-1 vector was a GE Healthcare Bio-Sci, Chicago, USA product. *Escherichia coli* (*E. coli)* DH10B F- end A1 recA1 galU galK deoR nupG rpsL ΔlacX74 Φ80lacZΔM15 araD139 Δ(ara,leu)7697 mcrA Δ (mrr-hsdRMS-mcrBC) Δ- was applied for cloning [30], while *E. coli* BL21 (DE3) F- ompT gal [dcm] [lon] hsdSB was used for protein expression [31].

### 3.2. Recombinant NColE7-KGNK Expression and Purification

The construction of the pGEX-6P-1-KGNK vector with an inserted R447G mutant of the nuclease domain of colicin E7 (NColE7) and the KGNK protein expression and purification with N-terminal Glutathione-S-Transferase (GST) fusion were described previously [20]. After the purification, the GST tag was cleaved according to the described method in [20]. The buffer of the purified product was exchanged to 20 mM N-2-hydroxyethylpiperazine-N-2-ethane sulfonic acid (HEPES) pH 7.7 using an Amicon ultra 15 mL centrifugal filter (Merck Millipore Ltd., Tullagreen Carrigtwohill, Co Cork IRL, Ireland). The purification steps of the protein were monitored via 15% (*w*/*v*) sodium dodecyl sulphate polyacrylamide gel electrophoresis (SDS PAGE) using a mixture of 116, 66.2, 45, 35, 25, 18.4 and 14.4 kDa unstained proteins as a marker (Thermo Scientific #26610). The experiment was carried out at 70 V for 30 min and then 120 V for 150 min using 0.1 M Tris-HCl, 0.1 M tricine and 0.1% (*w*/*v*) SDS, pH 8.3 cathode and 0.2 M Tris-HCl, and pH 8.9 anode buffers at RT. The resulting single band approved the homogeneity of the purified protein. The concentration of the purified protein was checked using UV absorbance spectrometry, recording the absorbance at 280 nm against the buffer baseline (Cary 8454 UV-Vis spectrophotometer, Agilent Technologies, Santa Clara, CA, USA). The molar absorbance of the protein used for the calculation of the concentration was estimated using an Expasy ProtParam tool [32] to be 12,490 M^−1^ cm^−1^. The circular dichroism spectrum was identical to that measured for the previously used batches of the protein.

### 3.3. Mass Spectrometric Experiments

Intact protein analysis was performed on an LTQ-Orbitrap Elite (Thermo Scientific) mass spectrometer coupled with a TriVersa NanoMate (Advion, Ithaca, NY, USA) chip-based electrospray ion source. Measurements were carried out in positive mode at 120,000 resolution in 2.5 mM ammonium hydrogen carbonate buffer (pH ~7.8). The protein concentration was 3.0 µM in each individual sample which contained various metal ions (added as ZnCl_2_, CuCl_2_, NiCl_2_ or CdCl_2_) at different molar ratios. Data evaluation, deconvolution to yield the masses of MH⁺ ions and spectrometric pattern simulations were performed using the Freestyle 1.6 or Xcalibur 2.2 software (Thermo Scientific). Analysing the data via the deconvolution of 10 × 10 consecutive scans we found that the standard deviation of the peak positions was less than 0.02 mass units. We did not make strict quantitative conclusions from the ESI MS spectra.

### 3.4. Circular Dichroism Spectroscopic Measurements

Circular dichroism (CD) spectra were recorded at room temperature utilising a Jasco J-1500 CD spectrometer using the following parameters. Wavelength range: 280–180 nm; path length: 0.2 mm (Jasco cuvette); D.I.T.: 2 s; bandwidth: 1.0 nm; scanning speed: 50 nm/min (continuous scanning mode); each spectrum was the average of three accumulated measurements. The concentration of the enzyme was 18.0 µM in 3–10 mM HEPES, pH 7.7. The measurements were carried out with apo enzyme, and in the presence of EDTA and/or various metal salts (ZnCl_2_, CuCl_2_, NiCl_2_ and CdCl_2_) at indicated molar ratios. Water and the buffer spectra were recorded for baseline correction, and the spectra were plotted without smoothing.

### 3.5. Catalytic Activity Assay

The catalytic activity of the KGNK mutant protein was monitored against plasmid DNA (pUC119) as a substrate. In a few experiments, the KGNK protein was treated with EDTA for comparison. The final concentration of the enzyme was 1.0 μM, while the DNA was 74 µM for base pairs in 20 mM HEPES, pH 7.7. The DNA cleavage was performed in the absence and presence of different metal ions of Zn^2+^, Cu^2+^, Cd^2+^ and Ni^2+^, at molar ratios of the enzyme to metal ion (1:1 and 1:3). The reaction mixtures were prepared by mixing all of the components, except for the plasmid DNA (pUC119) which was added to the mixture last. Then, it was incubated at 37 °C for various periods. Aliquots of 5 µL of the reaction mixture were taken four times and the reaction was stopped by adding 5 µL of 2% (*w*/*v*) SDS solution (1% (*w*/*v*) at final concentration) for enzyme denaturation. At least three replicates were carried out for each experiment.

The products of the DNA cleavage assays were checked via agarose gel electrophoresis (AGE). The products were run in 1% (*w*/*v*) agarose gel containing 0.5 µg/mL ethidium bromide for the visualisation of the DNA. Electrophoresis was performed in the TAE buffer (40 mM tris(hydroxymethyl) aminomethane, 20 mM acetic acid and 1 mM ethylenediaminetetraacetic acid, pH 8.0) using Bio-Rad wide mini sub cell VR GT, applying 7 V/cm potential gradients (100 V for 40 min). Gene Ruler 1 kb Plus DNA Ladder (Thermo Scientific) served as the reference, and non-cleaved plasmid DNA (pUC119) as a negative control was applied for comparison.

## 4. Conclusions

The R447G mutant of the nuclease domain of colicin E7 bacterial toxin was studied in this project in the presence of non-endogenous metal ions such as Ni^2+^, Cu^2+^ and Cd^2+^. The mutated protein with decreased activity allowed for easier monitoring of the catalytic process, but it also may have offered a new mechanism for DNA cleavage, since it lacks an important residue participating in DNA cleavage. This seems to be an intriguing question to answer.

Our circular dichroism and mass spectrometric results revealed that all of the metal ions used in this study bound to the active centre of the enzyme in the absence of Zn^2+^. However, they could not replace Zn^2+^ if it was already present in the active site. It was detected that the enzyme is very active in the presence of substoichiometric metal ion, which is difficult to remove via EDTA because of the competitive behaviour of the positively charged amino acid side-chains of the protein. It is also not unprecedented that a nuclease enzyme cleaves DNA in the absence of metal ions such as BfiI type II restriction endonuclease or the EDTA-resistant nuclease Nuc of Salmonella typhimurium [33,34]. On the other hand, we could see very high activity of the enzyme in the presence of Ni^2+^ ions, which could not be inhibited by the excess of the metal ion, but it was considerably inhibited by Zn^2+^, independent of the order of the metal ions added. These results suggest that the R447G enzyme cleaves DNA in a different manner from that of NColE7. We aim to deal with this exciting problem in a future project.

## Data Availability

Not applicable.

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
