# Peer review of "Hydrolytic Mechanism of a Metalloenzyme Is Modified by the Nature of the Coordinated Metal Ion"

_molecules, 2023, doi:10.3390/molecules28145511_

Round 1

Reviewer 1 Report

The present work focused on the interaction of several divalent metals (Ni2+, Cu2+ and Cd2+) with the nuclease domain of colicin E7. The results show that all metal ions interacted with the protein variant with a point mutation at R447 (R447G), but the presence of the metal had distinct effect on the catalytic activity of the enzyme. Although the topic is interesting and the results may contribute to the understanding of the catalytic mechanism of the enzyme and its regulation, I think some of the experiments/data are incomplete (or not properly presented) and the manuscript would benefit from a detailed revision. Therefore, in my opinion, the manuscript is not ready to be accepted for publication. The manuscript also would benefit with the addition of a Conclusions section, where the authors could highlight their findings and correlate with the hypotheses.

The following concerns should be addressed before the manuscript can be accepted for publication:

1. Introduction section. The introduction is very brief and does not introduce the system under study: 

1.1. The hypotheses driving this work should be clearly stated and substitution of a R residue per an G should be rationalized.

1.1. A tertiary structure of the protein should be presented, depicting the active site and the Im7 binding site, as well as the position of R447. 

1.3. What is the oligomeric state of the mutant protein in solution? Is it a dimer, as described for the WT nuclease domain?

1.4. How is the pI of the NColE7 and mutant proteins affected by the presence of the sequence left from the GST protease site relative to the nuclease domain without this tail? Does it affect the DNA binding?

1.5. In lines 46-47, the authors state that the complex NColE7-Im7 can be disrupted lowering the pH down to 3.0. Is the tertiary structure of the protein or the active form affected in this acidic condition?

2. Results and Discussion section.

2.1. Lines 78 to 89:  

2.1.1. What methods were used to verify the homogeneity, purity, and oligomeric state of the protein after purification, acidification at pH 3.0 and treatment with EDTA? Although SDS-PAGE is described to monitor the protein (section 3.2), no data is presented.

2.1.2. Is the protein produced fully bound to Im7 (one Im7 molecule per DNA binding site)? 

2.1.3. After acidification at pH 3.0 (no reference to this procedure in the Materials and Methods section) how was the Im7 counterpart removed?

2.2. Spectroscopic analysis using CD spectroscopy - Figures 1 and 2 data: 

2.2.1. To compare the intensity of spectra, I think it would be more appropriate to represent the molar ellipticity ([θ]) or delta epsilon (Δε).

2.2.2. I have trouble understanding how incubation with Zn2+ ions produce almost the same red shift as treatment with EDTA (which is negatively charged). In the legend, please specify the amount of EDTA added in the sample whose CD spectrum is identified as “P treated with EDTA”. I assume that this spectrum (“P treated with EDTA”) corresponds to that of the protein obtained after purification, treatment at pH 3.0 and with EDTA, while the “P” one refers to the protein prior to the EDTA treatment and after acidification at pH 3.0. Please clarify.

2.2.3. The change in intensity of the negative peak around 210 nm seems to vary quite a bit with Zn2+ (Figure 1b). Please comment.

2.2.2. Since the MS data shows that there are 2 populations of protein, 20% in the Zn-bound form, shouldn’t the CD spectrum of the protein (before addition of any metal) be slightly different when compared with the EDTA-treated protein one?

2.3. MS characterization – Figures 3 and 4:

2.3.1. Lines 131-133. What is the purity of the reagents used to prepare the protein samples? The presence of 20% of Zn-bound protein cannot be explained by a partial removal of the metal after treatment with EDTA?

2.3.2. The uncertainties associated with the measurements should be added in Table 1.

2.3.3. The sum of the areas under the major peaks in Figure 3d seems much larger than the remaining spectra.

2.3.4. In the spectra of the Cu2+-treated sample (Figure 4a), the smaller “KGNK-Zn(II)” peak does not seem present. Is this a different batch of protein?

2.4. Lines 178-179. What is the functional/structural significance of the shifting “toward the structure exhibiting lower charges” upon metal ions addition? Can one conclude that the open conformation is more active and stable?

2.5. Catalytic assays. How does the activity of the mutant protein (P only in Figure 7) compare with the wild type protein?

2.6. Based on the data, can the authors infer about the role of Ni2+? Is it a co-factor, an allosteric regulator or acts as a structural element?

3. Materials and Methods section.

3.1. As mentioned before some parts should be more detailed, namely section 3.2. 

3.2. How was the protein quantified by UV spectrophotometry? 

3.3. The number of replicates used in each assay should also be specified.

Minor questions:

1) Some examples of terms that that need to be revised: 

- Line 73 – “… frequent catalytic metal”; the authors may want to replace it by “… endogenous catalytic metal”

- Line 148 – Remove one of the “the”

- line 169 – Please correct “specis” to “species”

- Line 204 – I believe the authors meant “small” or “residual” instead of “minute”

- Line 259 – Instead of “order of the construction of the reaction mixture” , I suggest “order of addition of each component of the mixture” and instead of “in the opposite order” I suggest “in reverse order”

- Line 257 – Please replace “fo” by “of”

-       General questions:

-       Based on the data is Ni2+ ion a co-factor, an allosteric regulator or acts as a structural element?

-       Rationalize the substitution of a R per an G residue. The manuscript would benefit if a structural model of the mutant was included.

-       Does the mutation affects the oligomerization of the nuclease?

Some sentences are poorly constructed with problems with the 

vocabulary. A few examples are:

- "Zn2+ ion was shown to coordinate the purified NColE7 as its native metal ion.” In a metal complex, the metal is coordinated by ligands and in the case of metalloproteins, these mostly are side chain groups of amino acid residues and in some cases other molecules from the buffer, as water molecules.

- "applied metals ions" (to protein samples); "added" is a more appropriated term;

- "order of construction of the reaction mixture" instead of order of addition of each component of the reaction mixture;

- “The active centre of the enzyme is the C-terminal HNH motif with a characteristic ββα secondary structure topology. A divalent metal ion is bound here”;

- “a frequent catalytic metal ion”;

- “Most probably the Cu2+-bound protein was detected by MS”;

- “seemingly inhibiting”.

Reviewer 2 Report

Today more and more studies appear that prove the role of divalent metal ions in the process of regulating the activity of enzymes operating on DNA. The investigations oh Nafaee et al. is one of such studies. It devoted the interaction of nuclease domain of bacterial toxin, metalloprotein colicin E7 with Zn2+, Ni2+, Cu2+ and Cd2+  ions. The binding of these Me2+ was demonstrated by mass-spectrometry approaches. The DNA cleavage activity was also check. It depends from the nature of Me2+.

This work may be published. However, some aspects should be described more carefully to adapt the article for a wider range of scientists.

1. Introduction

1.1. The authors should characterize the full-length collicin E7 enzyme in more detail: its structure and specificity. From the Introduction it should be clear whether other domains of this protein affect the activity of the nuclease domain. Do they ensure the specificity of the enzyme. If so, which region does the full-length enzyme recognize in DNA?

1.2. It is required to explain in more detail and clearly, why it was necessary to use the mutant form of the protein under study.

2. Results and discussion

2.1. Both divalent metal ions and EDTA result in the same red-shift on the CD spectra. The authors should do a control experiment with a substance that is not able to interact with the enzyme, where such a shift would not be observed. The shift of 2 nm is so insignificant, at the level of experimental error, that, in my opinion, it is impossible to draw any conclusions about the binding of metal ions to a protein without independent control.

2.2. How many biological and technical repeats were done to create Fig. 4b? p-Values to estimate reliability of differences is necessary.

2.3. There are examples of other nucleases that cleave DNA in the absence of metal ions and are also found in the active site of histidine residues. The authors should have looked, е.g., through the early papers (2000-2005 years) of Prof. V. Siksnys concerning the metal-independent type IIS restriction enzyme BfiI and used these data in discussion. The comparison with EDTA-resistant nuclease Nuc of Salmonella typhimurium will be also useful.

2.4. How many repeats of analyses of NColE7 variant activity have been done? There sould be at least three converging repeats. Graphs with kinetic curves with error bars should be drawn for each case presented in Fig. 7-8.

3. The section Conclusion is necessary.

Minor correction

Line 217  Should be for the nuclease.

Round 2

Reviewer 1 Report

I commend the authors for considering all my comments and taking the opportunity to significantly improve the manuscript. The current version is, in my opinion, ready for publication.

 Minor editing of English language required.

Reviewer 2 Report

The authors improved the manuscript according to mу recommendations.

So, to my opinion, the paper could be published in present form.